# Risks of Biologic Therapy and the Importance of Multidisciplinary Approach for an Accurate Management of Patients with Moderate-Severe Psoriasis and Concomitant Diseases

**DOI:** 10.3390/biology11060808

**Published:** 2022-05-25

**Authors:** Ana Ion, Alexandra Maria Dorobanțu, Liliana Gabriela Popa, Mara Mădălina Mihai, Olguța Anca Orzan

**Affiliations:** 1Department of Dermatology, ‘Elias’ Emergency University Hospital, 011461 Bucharest, Romania; alexandramdorobantu@gmail.com (A.M.D.); liliana.popa@umfcd.ro (L.G.P.); mara.mihai@umfcd.ro (M.M.M.); olguta.orzan@umfcd.ro (O.A.O.); 2‘Carol Davila’ University of Medicine and Pharmacy, 020021 Bucharest, Romania

**Keywords:** chronic plaque psoriasis, biologic therapy risks, multidisciplinary approach

## Abstract

**Simple Summary:**

Psoriasis is a chronic multisystem inflammatory disease associated with a wide range of comorbidities including cardiovascular disease, hypertension, diabetes, hyperlipidemia, obesity, metabolic syndrome, anxiety, depression, chronic kidney disease, and malignancy. Currently available novel therapeutic options for moderate-severe psoriasis include tumor necrosis factor alpha inhibitors, inhibitors of the interleukin 17, and inhibitors of the interleukin 23. Apart from the concomitant diseases psoriasis patients may have, biologic therapy may cause significant complications requiring close collaboration between dermatologists and physicians of different specialties. Consequently, it was our main purpose to provide an overview of each class of biologic agents, as well as of the most frequent adverse events they may cause in psoriasis patients with concomitant diseases.

**Abstract:**

Psoriasis is a chronic multisystem inflammatory disease associated with a plethora of comorbidities including metabolic syndrome, cardiovascular disease, hypertension, diabetes, hyperlipidemia, obesity, anxiety, depression, chronic kidney disease, and malignancy. Advancement in unveiling new key elements in the pathophysiology of psoriasis led to significant progress in the development of biologic agents which target different signaling pathways and cytokines involved in the inflammatory cascade responsible for the clinical manifestations found in psoriasis. Currently available novel therapeutic options for moderate-severe psoriasis include tumor necrosis factor alpha inhibitors, inhibitors of the interleukin 17, and inhibitors of the interleukin 23. Nevertheless, concerns have been raised with respect to the possible risks associated with the use of biologic therapy requiring close collaboration between dermatologists and physicians of different specialties. Our aim was to perform an in-depth literature review and discuss the potential risks associated with biologic therapy in patients with psoriasis and concurrent diseases with a focus on the influence of novel therapeutic agents on liver function in the context of hepatopathies, particularly viral hepatitis. A multidisciplinary teamwork and periodic evaluation of psoriasis patients under biologic therapy is highly encouraged to obtain an accurate management for each case.

## 1. Introduction

Psoriasis is a chronic multisystem inflammatory disease marked by a strong genetic predisposition, as well as autoimmune pathogenic features [1]. Although it may vary between different regions, the worldwide prevalence of psoriasis is considered to be 2% [1]. It affects both men and women equally [1]. The hallmark of psoriasis is the excessive proliferation and altered differentiation of keratinocytes, which results in the typical clinical appearance of the lesions: well-demarcated erythematous plaques of variable size with silvery scale predominantly on the elbows, knees, lumbar region, and scalp (Figure 1), with the possibility of extensive involvement of the skin (Figure 2 and Figure 3) [2]. The most common morphologic variant of psoriasis is chronic plaque psoriasis, which accounts for up to 80% of cases; however, more severe clinical presentations, such as the erythrodermic variant, may be seen as well (Figure 4) [3]. Psoriasis not only affects the skin, but also the nails and joints [4]. At the present moment, it is well-recognized that psoriasis is associated with a plethora of comorbidities including metabolic syndrome, cardiovascular disease, hypertension, diabetes, hyperlipidemia, obesity, anxiety, depression, chronic kidney disease, and malignancy [4,5].

### 1.1. Key Elements in the Pathophysiology of Psoriasis

In the last years there has been significant advancement in acknowledging the complex pathophysiology of psoriasis [1]. It is believed that an excessive activation of certain components pertaining to the adaptative immune system is critical in the pathogenesis of the disease [3]. Initially, numerous cell types, including keratinocytes, macrophages, and natural killer T cells activate myeloid dendritic cells, which further secrete interleukin-12 (IL-12) and interleukin-23 (IL-23) [3]. IL-12 is responsible for the differentiation of naive T cells into T helper 1 (Th1) lymphocytes, while IL-23 is fundamental for the proliferation of T helper 17 (Th17) and T helper 22 (Th22) lymphocytes [3]. Th1 cells secrete tumor necrosis factor alpha (TNF-α) and interferon gamma ((IFN-γ), Th17 cells secrete interleukin-17 (IL-17), TNF-α, and interleukin-22 (IL-22), and finally, Th22 lymphocytes produce interleukin-22 (IL-22) [3]. IL-17 is produced under particular conditions such as environmental stimuli (pathogens or skin trauma) or due to exposure to keratinocyte autoantigens (such as LL37 cathelicidin or nucleic acid complexes), and its signals produce an inflammatory response of the feed forward type in keratinocytes, which further accelerates the development of psoriasis [6,7]. IL-17 and IL-22 activate STST3, leading to upregulation of the epidermal hyperplasia, a typical histopathologic finding in psoriasis [7]. Moreover, IL-17, along with TNF, promotes the transcription of proinflammatory genes (interleukin-1β, interleukin-6, interleukin-8, TNF) [8]. Consequently, the skin microenvironment is confronted with a plethora of inflammatory cells and mediators through a continuously positive feedback loop [5]. The predominant mechanism responsible for the pathogenic modifications in psoriasis is thought to be the IL-23-mediated activation of type 17 T-lymphocytes [9]. Although the entire pathophysiology of psoriasis may not be yet wholly unveiled, the above-mentioned insights led to outstanding progress concerning biologic therapy in moderate-severe psoriasis (Table 1) [10].

### 1.2. Biologic Therapy—A Revolutionary Therapeutic Option for the Management of Moderate-to-Severe Psoriasis

#### 1.2.1. Tumor Necrosis Factor Alpha (TNF-Alpha) Inhibitors

TNF-alpha inhibitors were the first biologic agents to be approved in the treatment of psoriasis [5]. Infliximab is an IgG1 murine–human antibody which has the ability to bind with high affinity and specificity to TNF-alpha, leading to important neutralizing and inhibitory activities which interfere with the inflammatory cascade in psoriasis [11]. The US Food and Drug Administration approved infliximab for the management of severe psoriasis in 2007 [12]. Various clinical trials have showed its efficacy regarding not only the appearance of the skin, but also the overall health-related quality of life [13,14,15]. Adalimumab is a humanized monoclonal antibody with anti-TNF-alpha activity which was firstly used for the management of rheumatoid arthritis [16]. A clinical trial from 2006 by Gordon KB et al. aimed to establish the clinical response to adalimumab in 147 patients with moderate-to-severe psoriasis [17]. The first group received 40 mg adalimumab every other week, the second group received adalimumab every week, while the third group represented the placebo group [17]. Results showed that at week 12, 53% of the patients with biologic therapy every other week, 80% of patients taking adalimumab on a weekly basis, and 3% of the placebo group could achieve 75% improvement in Psoriasis Area and Severity Index (PASI) score, and these responses were sustained for 60 weeks [17]. In 2011, a 16-week clinical trial by Strober BE et al. demonstrated that adalimumab was a suitable biologic agent for patients with suboptimal response to etanercept, in a study for which 162 patients with prior etanercept, methotrexate, or narrowband ultraviolet B phototherapy were enrolled. At the end of the 16 weeks, Physician Global Assessment of “clear” or “minimal” was achieved by 52% of all enrolled subjects and by 49% of the etanercept subgroup [18]. Certolizumab pegol is a PEGylated, Fc-free TNF-alpha inhibitor whose efficacy is supported by phase III clinical trials, particularly two multicentered, randomized, double-blind, placebo-controlled studies: CIMPASI-1, which included 234 adults, and CIMPASI-2, which included 227 adults, who were randomly assigned to receive either 400 mg certolizumab pegol every two weeks, 200 mg certolizumab pegol every two weeks after loading dose of 400 mg, or placebo every two weeks [19]. Results of CIMPASI-1 showed that PASI 75 was achieved by 76% of patients in the first group, 67% of patients in the second group, and 7% of adults in the placebo group, while in CIMPASI-2, PASI 75 was achieved by 83% of patients in the first group, 81% percent of adults in the second group, and 12% of patients in the third group, respectively [19]. Certolizumab pegol is also ideal for the management of psoriasis in pregnant women, since it does not bind to the neonatal Fc receptor for IgG, showing a minimal transplacental transfer [20]. Etanercept is another tumor necrosis factor alpha inhibitor approved by the US Food and Drug Administration for the use in children of four years of age and older and for patients with psoriatic arthritis [21]. A randomized, double-blind, placebo controlled clinical trial from 2006 by Tyring S et al. showed that 47% (147 of 311) patients who received 50 mg twice weekly of etanercept achieved PASI 75 at week 12 compared to only 5% (15 out of 306) of those receiving placebo [22].

#### 1.2.2. Inhibitors of the Interleukin-17 (IL-17) Pathway

Secukinumab is a monoclonal antibody, an inhibitor of the IL-17 pathway effective for both moderate-severe psoriasis and psoriatic arthritis [23]. In 2014, Langley RG et al. provided results of two phase III double-blind, 52-week, clinical trials [24]. The first one, ERASURE, sought to assess the efficacy and safety of two fixed therapeutical regimens with secukinumab in psoriasis and included 738 patients who were administered a 300 mg or 150 mg subcutaneous dose of secukinumab once weekly for five weeks, then every four weeks [24]. The primary objective of the study was to demonstrate the superiority of secukinumab over placebo at week 12 concerning PASI 75 [24]. Results were promising, showing that at week 12, the proportion of subjects who met the criteria for PASI 75 was higher for the secukinumab groups than for the placebo group: 81.6% for the 300 mg secukinumab group, 71.6% for the 150 mg secukinumab group, and 4.5% for the placebo group [24]. The second clinical trial, FIXTURE, compared secukinumab with etanercept in terms of efficacy for a whole year period [24]. It included 1306 patients who either received secukinumab 300 mg or 150 mg subcutaneous for five weeks, then every four weeks or etanercept 50 mg subcutaneous twice weekly for 12 weeks, then once weekly [24]. At week 12, PASI 75 was achieved by 77.1% of the 300 mg secukinumab group, 67.0% of the 150 mg secukinumab group, 44.0% of the etanercept group, and 4.9% of the placebo group [24].

Ixekizumab is a human monoclonal antibody directed against IL-17A approved for the management of moderate-severe psoriasis in adults and of psoriatic arthritis [25]. Clinical studies demonstrating its efficacy are represented by the UNCOVER trials [26]. In 2015, Griffiths CE et al. published the results of two phase III randomized, double-blind, multicenter clinical trials: UNCOVER-2 and UNCOVER-3, which compared the efficacy of ixekizumab to that of etanercept or placebo [26]. UNCOVER-2 included 1224 patients who were then randomly assigned to be administered either placebo (*n* = 168), etanercept (*n* = 358), or ixekizumab every two weeks (*n* = 351) or every four weeks (*n* = 347) [26]. For the patients receiving ixekizumab every two weeks, PASI 75 was achieved at 12 weeks by 89.7%, while for the patients receiving ixekizumab every four weeks, the proportion was 77.5% [26]. The UNCOVER-3 study included 1346 patients who were also randomly assigned to receive either placebo (*n* = 193), etanercept (*n* = 382), or ixekizumab every two weeks (*n* = 385) or every four weeks (*n* = 386) [26]. As for UNCOVER-2, results were astonishing, with PASI 75 at week 12 being achieved by 87.3% of the patients from the group that received ixekizumab dosing every two weeks and by 84.2% of the patients from the group the received ixekizumab dosing every four weeks [26]. Concerning the response to etanercept, 149 (41.6%) patients from UNCOVER-2 and 204 (53.4%) patients from UNCOVER-3 achieved PASI 75 at the end of the 12-week period [26].

Brodalumab is yet another monoclonal antibody with anti-IL-17A activity approved for the treatment of moderate-severe psoriasis in those who are adequate candidates for phototherapy or systemic therapy and have either lost response or failed to respond to systemic therapy [27].

#### 1.2.3. Inhibitors of the Interleukin-23 (IL-23) Pathway

Another important category of biologic therapy is represented by the inhibitors of the IL-23 and the related cytokines. Ustekinumab is a biologic agent which targets both IL-12 and IL-23 [28]. In 2008, Leonardi CL et al. published the results of a 76-week phase III randomized, placebo-controlled, double-blind clinical trial (PHOENIX-1) on 766 patients who received either 45 mg (*n* = 255) or 90 mg (*n* = 256) ustekinumab at weeks 0 and 4, and then every 12 weeks or placebo (*n* = 255) [28]. At week 12, results showed that PASI 75 was successfully achieved by 67.1% of patients receiving 45 mg of ustekinumab and 66.4% of patients receiving 90 mg of ustekinumab, thereby fulfilling the primary endpoint of the study [28]. Ustekinumab has proven to be safe and effective in adolescents affected by psoriasis, as shown by the CADMUS phase III study from 2015 by Landells I et al., with responses at week 0, 4, and 12 similar to those found in adults [29].

Guselkumab is another therapeutic agent which targets the p19 subunit of the IL-23 and IL-39, being a human immunoglobulin G1 monoclonal antibody. Two phase III, 48-week randomized, double-blind, active comparator- and placebo-controlled trials, VOYAGE 1 and VOYAGE 2, respectively, have demonstrated the superiority of guselkumab when compared to adalimumab or placebo for the treatment of patients with moderate and severe psoriasis [30,31]. In 2019, Reich K et al. compared the efficacy of guselkumab versus secukinumab regarding the treatment of moderate-to-severe psoriasis in a phase III, randomized study—the ECLIPSE trial [32]. Of 1048 patients enrolled, 534 received guselkumab and 514 received secukinumab [32]. At week 48, PASI 90 was achieved by 84% of the guselkumab group and by 70% of the secukinumab group, showing that the former biologic agent demonstrated a superior long-term efficacy [32].

Tildrakizumab is a human immunoglobulin G1 monoclonal antibody, which binds to the p19 subunit of the IL-23, being approved for the treatment of moderate-to-severe psoriasis in adults [33]. There are two phase III randomized clinical trials, reSURFACE 1 and reSURFACE2, from 2017, by Reich K et al., which prove the superiority of tildrakizumab versus placebo or etanercept [33]. Thus, tildrakizumab 200 mg and 100 mg were well-tolerated and efficacious for the management of moderate-to-severe plaque psoriasis compared to placebo or etanercept [33]. 

Risankizumab is a human monoclonal antibody with activity against the p19 subunit of IL-23 and IL-39 which is approved for the treatment of moderate and severe psoriasis in adults [34]. A recent phase III, multicenter, open-label, randomized clinical trial from 2021 by Warren RB et al., the IMMerge study, compared the efficacy and safety of risankizumab versus secukinumab over a 52-week period [34]. The trial included 327 patients receiving either risankizumab (164 patients), 150 mg at weeks 0,4, and then every 12 weeks or secukinumab (163 patients), 300 mg at weeks 0, 1, 2, 3, 4, and then every four weeks [34]. At week 52, PASI 90 was achieved by 86.6% of patients receiving risankizumab and by 57.1% of patients receiving secukinumab [34], showing the superiority of risankizumab in terms of efficacy and safety, with far less-frequent dosing compared to the latter biologic agent [34]. In 2019, a randomized, double-blind, active-comparator-controlled phase III clinical trial by Reich K et al. compared the efficacy and safety of risankizumab versus adalimumab in adult patients with moderate-to-severe psoriasis—the IMMvent trial [35]. The study included 605 patients who received either risankizumab (*n* = 301) 150 mg subcutaneously at weeks 0 and 4 or adalimumab (*n* = 304) 80 mg subcutaneously at the moment of randomization, then 40 mg at weeks 1, 3, and 5, then every other week [35]. Results showed that at week 16, PASI 90 was successfully achieved by 72% of the patients from the risankizumab group and by 47% of patients from the adalimumab group [35]. At the 16th week, the intermediate responders to adalimumab were re-randomized to either switch to 150 mg risankizumab or continue with 40 mg adalimumab until the 44th week, when results showed that PASI 90 was achieved by 66% of patients switching to risankizumab and by only 21% of patients continuing adalimumab [35].

## 2. Material and Methods

An in-depth literature review regarding the main risks and complications of biologic therapy currently available for the management of moderate-to-severe psoriasis was developed. The interrogation was performed through PubMed using the following keywords: ‘chronic plaque psoriasis’, ‘biologic therapy risks’, ‘multidisciplinary approach’. It included papers published between 2006 and 2021.

## 3. Complications of Biologic Therapy and the Importance of a Multidisciplinary Approach in the Management of Psoriasis Patients

### 3.1. Main Complications of Biologic Therapy

As previously described, the perspective of biologic therapy in psoriasis and the advancement in the field of novel therapeutic agents greatly improved the approach to moderate-to-severe psoriasis. Nevertheless, concerns have been raised with respect to the possible risks associated with the use of biologic agents (Figure 1). In Table 2, we clearly summarize which biologic therapy will likely be effective and which agent would rather be contraindicated. (Table 2). In the future, through the field of pharmacogenomics, it may be possible to precisely identify which biologic agent may cause adverse reactions to each patient using biomarkers [36,37]. Until this advanced technique is available in clinical practice, it is of outmost importance to carefully monitor each case once the biologic therapy is initiated in order to avoid complications. 

It is believed that infection represents one of the main reasons for discontinuation of biologic therapy [38]. A recent French cohort study from 2021 by Penso L et al. assessed the risk of serious infections in patients with psoriasis under biologic therapy and apremilast, a phosphodiesterase-4 inhibitor, with etanercept being the comparator [39]. A serious infection is defined as requiring intravenous antibiotic therapy or resulting in hospitalization or death [39]. There were 44239 new users of biologic therapy included, with a median follow-up of 12 months, data being obtained from national healthcare records [39]. The authors found the following results: gastrointestinal infections were the most frequent serious infections—38.9% (645 patients) [39]. The risk of serious infections was superior for patients taking infliximab (weighted hazard ratio [wHR] 1.79, 95% CI 1.49–2.16) and adalimumab (wHR 1.22, 95% CI 1.07–1.38) and lower for patients under ustekinumab (wHR 0.79, 95% CI 0.67–0.94) versus etanercept. No increased risk was found with ixekizumab, secukinumab, brodalumab, guselkumab, or apremilast versus etanercept, nor was risk increased for new users of apremilast or the IL-23 inhibitor guselkumab versus etanercept [39]. Another cohort study from 2021 by Jin Y et al. on 123383 patients with psoriasis or psoriatic arthritis compared the risk of infection requiring hospitalization of patients initiating ustekinumab versus apremilast, adalimumab, certolizumab, etanercept, golimumab, ixekizumab, or secukinumab between 2009 and 2018 [40]. It seemed that patients under ustekinumab had a risk of serious infections 1.4 to 3 times lower compared to those taking the other studied agents [40]. In 2019, a study from the British Association of Dermatologists Biologics and Immunomodulators Register (BADBIR) showed that infliximab was associated with an increased risk of serious infection when compared to non-biologic systemic therapies [41]. In 2015, Kalb RE et al. published the results from the Psoriasis Longitudinal Assessment and Registry (PSOLAR) between 2007 and 2013 on the risk of serious infection in patients with psoriasis under systemic or biologic therapy [42]. Data from 11466 patients were analyzed: the cumulative incidence rate of serious infection was that of 1.45 per 100 patient-years (*n* = 323) [42]. Commonly reported types of serious infections were pneumonia and cellulitis [42]. The rates of serious infection were 2.49, 1.97, 1.47, 0.83, per 100 patient-years for the infliximab, adalimumab, etanercept, and ustekinumab group, respectively [42]. Consequently, the results from the register show that the risk for serious infections is higher with infliximab and adalimumab compared to nonbiologic therapies, while there was no increased risk for ustekinumab or etanercept [42]. Certain constitutive and external factors were associated with an increased risk of serious infections: age, smoking, a history of diabetes mellitus or serious infection, as well as infliximab or adalimumab exposure [42].

The use of biologic agents with anti-IL17 activity has been correlated with the appearance of fungal infections, since IL-17 is required for proper immunological protection against fungal pathogens [43]. In a systematic review from 2017 by Saunte DM et al. in which anti-IL-17 trials were compiled, the authors found that candidal infections were present in 4.0% of patients treated with brodalumab, in 3.3% of those treated with ixekiumab, and in 1.7% of patients receiving with secukinumab, versus 2.3% of those under ustekinumab and 0.8% of those receiving etanercept [43]. These findings show that in those patients affected by psoriasis where fungal infections represent a concern, other groups of biologic agents may successfully be taken into consideration [43]. 

The possibility of tuberculosis acquisition or reactivation is an important matter of concern with biologic therapies, especially with TNF-alpha inhibitors [44,45]. Clinical practice guidelines advise screening for latent tuberculosis using interferon gamma release assay and a plain chest radiograph in order to rule out abnormal modifications at baseline, before considering initiation of a biologic agent for the management of psoriasis [44,45]. For those patients requiring therapy for latent tuberculosis, at least two months of specific treatment should be completed prior to biologic therapy initiation [45]. However, it is recommended to carefully weight the risk-benefit of initiating specific chemoprophylaxis, since anti-tuberculous therapy is associated with important adverse reactions [46]. 

Due to the increased risk of infection with some biologics, hypotheses concerning a possible attenuation of the immune response in patients with psoriasis under biologic therapy against the severe acute respiratory syndrome coronavirus-2 (SARS-CoV-2) have been raised at the beginning of the pandemic. A retrospective observational study from August 2020, conducted by Gisondi et al. on 5206 patients with psoriasis on biologic therapy, found through patient records that apparently, there were no COVID-19-related deaths compared to an incidence rate of 1.6 in the general population [47]. A major limitation of this study was the absence of a control group [48]. An observational study from 2020 by Mahil SK et al. aimed to determine the course of COVID-19 infection in psoriasis patients and to identify those factors possibly associated with hospitalization. The study included 374 clinician-reported psoriasis patients with confirmed/suspected COVID-19 infection, of which 71% were receiving biologic therapy, 18% were receiving a nonbiologic therapy, while 10% were under no treatment for psoriasis. Results showed that 93% (*n* = 348) of patients fully recovered from the COVID-19 infection, 21% (*n* = 77) required hospitalization, and 2% (*n* = 9) died [49]. Hospitalization was more frequently found in patients using non-biologics than in those using biologic therapy (with an odd ratio of 2.84 CI = 1.31–6.18) [49]. However, there were no significant differences found between classes of biologic agents [49]. Age, male sex, and preexistent chronic lung disease were risk factors associated with higher hospitalization rates [49].

The risk of malignancy is another issue of concern when using biologic therapy for the treatment of moderate-to-severe psoriasis. Nonetheless, most biologic agents are relatively new, therefore there is still a need for surveillance and data collection from registries to establish a potential correlation between long-term biologic therapy exposure and cancer development. A case-control study from 2018 by Doval Garcia I et al. aimed to determine a dose–response relationship between long-term exposure to biologic agents and risk of cancer [50]. The study included 728 subjects who had developed a first cancer, and for each patient, there were four cancer-free controls [50]. Results showed that the overall risk of a first cancer was not correlated with a lengthy exposure to biologic therapy and this was also the case for squamous and basal cell carcinomas [50]. In order to minimize any potential oncologic risk, clinical guidelines advice for psoriasis patients to participate in screening programs for cancer and suggest both physicians and patients to carefully consider the risks and benefits of biologic therapy discontinuation [44,45]. 

Cardiovascular risk in patients receiving biologic therapy mostly refers to heart failure (HF) and major cardiovascular events (MACE) [48]. It is believed that TNF-alpha inhibitors may have deleterious cardiovascular effects; however, data from literature provides rather conflicting results [51]. Nevertheless, the risk of MACE in psoriasis patients with biologic therapy has been studied [48]. A prospective cohort study from 2020 by Rungapiromnan W et al. evaluated the risk of acute coronary syndrome, myocardial infarction, unstable angina, and stroke in patients with chronic plaque psoriasis under adalimumab, etanercept, or ustekinumab [52]. The study included 5468 biologic naive patients, of which 3204 were exposed to adalimumab, 1313 to etanercept, and 951 to ustekinumab [52]. Subsequently, on a secondary analysis, the study also included 2189 patients who received methotrexate [52]. The median follow-up was two years [52]. Results showed no difference of risk in terms of major cardiovascular events between biologic agents: ustekinumab versus adalimumab, etanercept versus adalimumab, or between methotrexate and adalimumab [52]. In 2019, Deodhar A et al. published a pooled clinical trial and some post-marketing surveillance data concerning the long-term safety of secukinumab when used by patients with moderate-to-severe psoriasis, psoriatic arthritis, and ankylosing spondilitis [53]. The authors showed that the risk of MACE for patients with psoriasis receiving secukinumab was rather low (0.3 exposure adjusted incidence rates per 100 patient-years) [53].

The recommendations provided by clinical practice guidelines state that TNF-alpha inhibitors are relatively contraindicated in patients with congestive heart failure [44,45]. In the event of worsening or newly diagnosed heart failure, biologic therapy with TNF-alpha inhibitors should be stopped and alternative biologic agents should be taken into consideration [44,45].

In the last years, a range of rather paradoxical inflammatory adverse effects to biologic therapy has been described, such as cutaneous disorders, inflammatory bowel disease, or interstitial lung disease. The cutaneous adverse effects vary greatly, from eczema and pustular and lichenoid eruptions to paradoxical psoriasis, as shown by Garcovich S et al. in a review from 2019 regarding the skin reactions of patients with rheumatologic disorders under biologic therapy [54]. In the review, 46% of the cases had a personal history of atopy, which may predispose to cutaneous adverse effects [54]. A recent publication from 2021 by Brunner PM et al. described the eczematous reactions observed in patients with psoriasis treated with ustekinumab, ixekizumab, and etanercept and reported that a prior history of eczema may represent a predisposing factor for the appearance of cutaneous adverse reactions in this particular category of patients [55]. As for inflammatory bowel disease, it seems that IL-17 inhibitors may flare this condition [56,57]. Clinical trials of IL-17 inhibitors for the treatment of psoriasis, psoriathis arthritis, ankylosing spondilitis, and rheumatoid arthritis--namely ixekizumab, secukinumab, and brodalumab—resulted in newly diagnosed inflammatory bowel disease secondary to the use of these agents [58,59].

### 3.2. Laboratory Parameter Dynamics and Paraclinical Adverse Events of the Liver Function in Patients with Psoriasis under Biologic Therapy

Biologic therapy, as a novel therapeutic approach for moderate-to-severe psoriasis in adults, requires periodic careful monitoring through routine laboratory tests, which mostly include complete blood count, inflammation markers, and evaluation of the liver and renal function, to prevent any modification of the evaluated parameters. We will further describe data found with respect to laboratory parameters in patients with psoriasis under biologic therapy, with a focus on altered liver function tests. In 2020, Ataseven A et al. conducted a clinical study in which they compared TNF-alpha and interleukin inhibitors treatments in patients with psoriasis with respect to laboratory dynamics of the assessed parameters [60]. The study included 191 patients with psoriasis who had received treatment for more than six months with either infliximab, adalimumab, etanercept, ustekinumab, or secukinumab, data being collected between 2013 and 2020 [60]. Routine laboratory parameters were evaluated before the initiation of the biologic treatment, at three months, then at a final evaluation and included hemoglobin (Hgb), red blood cells (RBC), serum levels of creatinine, c-reactive protein (CRP), aspartate aminotransferase (AST), and alanine aminotransferase (ALT), among others [60]. Results showed that AST, ALT, and creatinine had higher values in the TNF-alpha inhibitors group compared to the interleukin inhibitors group [60]. Factors increasing the value of both AST and ALT included the use of additional drugs: acitretin (six patients in the TNF-alpha inhibitors group), methotrexate (8 patients in the TNF-alpha inhibitors group), and isoniazid (44 patients in the TNF-alpha inhibitors group) [60]. The effect of concomitant drug use on AST and ALT was statistically significant (*p* < 0.001) [60]. Moreover, CRP values were reduced significantly in the TNF-alpha inhibitor group compared to the interleukin inhibitor group [60]. The authors concluded that TNF-alpha inhibitors may be more effective in reducing inflammation, while IL inhibitors may be safer in terms of biochemical parameters [60]. However, modifications in biochemical markers were largely due to concomitant drug use [60]. 

In 2019, Gerdes S et al. evaluated the course of liver and metabolic parameters in patients affected by psoriasis under secukinumab, etanercept, and placebo, with a focus on the former biologic agent [61]. Three randomized controlled trials, SCULPTURE, FIXTURE, and ERASURE were included in the pooled analysis [61]. Patients received secukinumab either 300 or 150 mg, etanercept 50 mg in the FIXTURE trial or placebo [61]. Among the assessed parameters there were AST and ALT [61]. Results showed that after a 52-week period of therapy, patients under secukinumab showed stable levels of AST and ALT, while patients under etanercept had an increase in their level of liver enzymes, starting from week 16 [61]. This finding is consistent with other data from the literature concerning the elevation of liver enzymes in patients under biologic therapy with etanercept [62]. 

A retrospective analysis from 2017 by Hoffmann J et al. Evaluated some routine laboratory parameter modifications in patients with psoriasis on long-term therapy with adalimumab, etanercept, and ustekinumab [63]. Laboratory tests included complete blood count, hemoglobin, CRP, lactate dehydrogenase (LDH), transaminases, gamma-glutamyl transferase (GGT), triglycerides, total cholesterol, blood urea nitrogen (BUN), and creatinine levels measured at baseline and during routine evaluation [63]. The laboratory adverse events were classified according to the Common Terminology Criteria for Adverse Events (CTCAE) [63]. Findings showed a dynamic alteration of laboratory parameters rather early during treatment, with 11 out of 15 CTCAE laboratory adverse events including elevated liver enzymes [63]. However, the alterations were either self-limited or due to simultaneous drug use, particularly methotrexate, or due to psychological conditions, such as alcohol intake [63]. Consequently, the authors recommend proper screening of liver function according to each patient’s comorbidities and concomitant therapy [63]. 

In case of unexplained elevation in AST and/or ALT in a patient with psoriasis receiving biologic therapy, the British Association of Dermatologists guidelines suggest retesting for viral hepatitis: surface antigen and core antibody for hepatitis B and immunoglobulin G for hepatitis C [45]. Nowadays, there is still controversy regarding the use of biologic therapy in patients with moderate-to-severe psoriasis who have concomitant hepatitis B infection since the safety of biologic agents in this particular group of patients is yet not thoroughly established [64]. A prospective study from 2018 by AlMutairi N and Abouzaid HA on the safety and efficacy of biologic therapy in patients with psoriasis and concurrent chronic viral hepatitis included 39 patients with chronic inactive or occult hepatitis B and chronic hepatitis C which received biologic therapy for at least 24 weeks [64]. Viral markers, viral DNA load and liver enzymes were evaluated at baseline, during the period of treatment with biologic agents and at the end of the follow-up period [64]. Results showed that patients with positive HBV and HCV markers did not show viral reactivation and the levels of AST and ALT were within the normal range throughout the assessment period [64]. In 2013, Navarro R et al. conducted a retrospective, multicenter study to assess both the safety and the effectiveness of TNF-alpha inhibitors and ustekinumab in patients with psoriasis and chronic viral hepatitis B or C [65]. The study encompassed 20 patients with hepatitis C virus and five with hepatitis B virus who had received therapy with at least one biologic agent: there were 21 treatments with etanercept, four with adalimumab, two with infliximab, and four with ustekinumab [65]. During the study, clinical data, laboratory data, and imaging were recorded [65]. For those with hepatitis C, results showed that AST, ALT, and GGT doubled their value from baseline only in one patient, who was receiving etanercept, two other cases presented with increased viral load during their follow-up period, and two patients who were receiving etanercept were diagnosed with hepatocellular carcinoma [65]. For those with hepatitis B, there were no significant alterations in viral load or liver enzymes [65]. All subjects managed to achieve PASI 75 during follow-up [65]. The authors concluded that biologic therapy is safe and effective for patients with psoriasis and concurrent viral hepatitis; however, there might be a risk of aggravation and/or reactivation which should be carefully taken into consideration [65]. A multicenter study from 2015 by Bueno-Sanz J et al. estimated the risk of reactivation of hepatitis B virus infection in psoriasis patients treated with biologic therapy by making a retrospective analysis of 20 cases from the Spanish Registry of Adverse Events Associated with Biologic Drugs in Dermatology (BIOBADADERM) [66]. There were 20 patients treated with at least one biologic agent and serologic evidence of past hepatitis B virus infection, which is negative hepatitis B surface antibody and positive total hepatitis B core antibody and at least one biologic included [66]. The clinical manifestations, serologic variables, and liver enzymes were collected before, during, and at the end of a 40-week follow-up [66]. At the end of this period, the viral load was assessed for all patients [66]. Results showed that at the end of the follow-up period, there were no important alterations concerning the values of ALT and the criteria for hepatitis B virus reactivation were not fulfilled by any patient [66]. Although reactivation did not happen in this particular case series, it is critical to measure the viral load in a patient with a history of hepatitis B virus infection prior to the initiation of biologic therapy due to the potentially severe complications associated with hepatitis B virus reactivation [66]. 

A review from 2019 by Poelman SM on an update regarding practical guidelines for patients affected by psoriasis on biologics provides valuable insight on the approach to patients with psoriasis and concomitant viral hepatitis [67]. In patients with a history of hepatitis B virus infection, the evaluation of the viral load prior to the initiation of biologic therapy becomes crucial due to the potentially severe complications of the reactivation [67]. Moreover, high-risk patients should have the following tests performed before the initiation of biologic therapy: antibody to hepatitis B core antigen (antiHBc), hepatitis B surface antigen (HBsAg), antibody to hepatitis B surface antigen (anti-HBs) [67]. Current guidelines advise for a collaboration between dermatologists and hepatologists for chronic carriers of hepatitis B virus [67]. As for hepatitis C virus infection, screening with anti-hepatitis C virus antibodies is strongly recommended; in the case of active infection, antiviral therapy should be administered prior to biologic therapy [67].

A recent review from 2021 by Thatiparthi A et al. suggests an algorithm for the management of patients with moderate-to-severe psoriasis on biologic therapy with comorbid conditions [68]. Regarding the concurrent hepatitis B virus infection in patients with psoriasis under biologic therapy, the authors provide data from several clinical studies from the last eight years in order to clarify which biologic agent may be more suitable for this special population. The expert opinion algorithm the authors propose includes IL-17 inhibitors as a first option and IL-23 inhibitors as a second therapeutic option for patients with psoriasis and concomitant hepatitis B virus infection, while TNF-alpha inhibitors should be avoided [68]. A retrospective cohort study from 2017 by Snast I et al. showed that the yearly reactivation rates were higher in those patients with a chronic hepatitis B virus infection (13.92%) on TNF-alpha inhibitors than for patients with resolved infection (0.32%) [69]. In a multicenter study from 2021 by Chiu HY et al., the authors identified those predictors of hepatitis B and C reactivation in patients with psoriasis on biologics: the absence of antiviral prophylaxis (*p* = 0.046), HBsAg-seropositivity (*p* < 0.0001), hepatitis B e-antigen positivity (*p* = 0.0134) [70]. Moreover, it was found that the risk of reactivation was higher with TNF-alpha inhibitors than with IL-17 inhibitors [70]. A prospective study from 2018 by Chiu HY et al. on 49 patients with psoriasis on secukinumab and concomitant hepatitis B virus infection with positive HBsAg found that this category had a higher risk of reactivation than HBsAg-negative and HBcAg-positive patients (24.0 vs. 4.17%, *p*-value = 0.047) [71]. Ustekinumab seemed both safe and efficacious when used in patients with psoriasis and hepatitis B with antiviral prophylaxis and adequate monitoring, as shown by two retrospective cohort studies [72,73]. A prospective cohort study from 2018 by Ting SW, Chen YC, Huang YH on 93 patients with hepatitis B virus infection and psoriasis on ustekinumab showed that inactive HBV carriers had a reactivation rate of 17.4% in the absence of antiviral prophylaxis compared to no cases when receiving antiviral prophylaxis [74].

### 3.3. Multidisciplinary Collaboration as a Key Step for the Appropriate Management of Psoriasis Patients under Biologic Therapy

Psoriasis is a systemic, chronic, inflammatory condition associated with a wide range of comorbidities, namely psoriatic arthritis, non-alcoholic fatty liver disease, inflammatory bowel disease, cardiovascular risk (metabolic syndrome, obesity, hypertension, hyperlipemia), chronic kidney disease, and psychological disorders such as anxiety and depression [75]. Concerning psoriatic arthropathy, a recent interview study from 2021 by Sumpton Daniel et al. on the perspective of the clinicians of a shared care of psoriatic arthritis and psoriasis between dermatology and rheumatology found that shared care models may improve the accuracy of management by making the patient the center of care [76]. In 2018, a position statement on the management of psoriasis patients with concurrent diseases was published [75]. The statement was developed by a board of ten experts from different specialties (four dermatologists, one rheumatologist, one cardiologist, one gastroenterologist, one nephrologist, one endocrinologist, and one psychiatrist) [75]. The objective was to provide the dermatologist with an accurate tool for systematizing the diagnosis of psoriasis comorbidities, thereby facilitating the process of decision-making about the referral and the management of patients with psoriasis and concomitant diseases [75]. 

Biologic therapy presents new safety issues to be taken into consideration in different populations affected by psoriasis: demyelination, chronic viral infections, cardiovascular outcome, lymphoma, tuberculosis, pregnancy, surgery, and vaccination, to name a few [77]. A review from 2020 by Bernadini N et al. emphasizes the importance of both careful screening for infectious diseases in psoriasis patients candidates to begin biologic therapy, as well as on the close collaboration with the infectious diseases physician in patients with personal history of tuberculosis, human immunodeficiency virus infection or chronic hepatitis B or C virus infection [78]. The authors advise active collaboration with the infectious diseases physician for a successful approach of the psoriasis patient with infectious comorbidities [78]. In 2019, a review by Olveira A, Herranz P, and Montes ML on the cooperation between dermatologists and hepatologists regarding the approach of patients affected by psoriasis hepatopathies was published [79]. In this paper, the authors recommended that hepatologists and dermatologists should work together to ensure the evaluation of the optimal therapy for each patient, selecting the therapeutic option that may be suitable for both psoriasis and the liver disease [79]. Moreover, the severity of each case should be carefully analyzed and hepatotoxic drugs avoided, if possible [79]. The interdisciplinary collaboration between physicians of different specialists is of utmost importance to assure the proper management of the patients affected by psoriasis.

## 4. Discussion

Psoriasis is a chronic systemic inflammatory disease associated with numerous comorbidities ranging from metabolic syndrome and cardiopathy to psoriatic arthritis and psychological disorders [4,5,75]. Apart from the concomitant diseases psoriasis patients may suffer from, biologic therapy may cause significant complications; therefore, finding the right biologic agent for each patient may prove challenging [43,44,45,46,48,50,51,52,54,55,56,57,64,65,72]. It seems that a promising tool for establishing which biologic agent is most suitable for each patient is represented by pharmacogenomics. In the future, particular biomarkers may be tested for each case in order to identify which biologic agent will likely be effective and which will cause complications [36,37]. Nevertheless, at the present moment, the key step for the proper management of psoriasis patients under biologic therapy includes the close collaboration between dermatologists and colleagues of other specialties [75,76,77,78,79]. For that reason, it was our main purpose to provide an overview on each class of biologic agents, as well as on the most frequent adverse events they may cause in psoriasis patients with concomitant diseases.

## Data Availability

Not applicable.

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
