# Peer review of "Risks of Biologic Therapy and the Importance of Multidisciplinary Approach for an Accurate Management of Patients with Moderate-Severe Psoriasis and Concomitant Diseases"

_biology, 2022, doi:10.3390/biology11060808_

Round 1

Reviewer 1 Report

The work "Risks of biologic therapy and the importance of multidisciplinary approach for an accurate management of patients with moderate-severe psoriasis and concomitant diseases" is of undoubted interest. 
The authors conducted an in-depth review of the literature and discussed the potential risks associated with biological therapy in patients with psoriasis and comorbidities.
The work lacks some practical conclusion.
What risks can each of the biological agents provoke? In what comorbidities, which biological agent will be effective and in what comorbidities, which biological agent will be contraindicated?
This information can be presented graphically or as a table.
It is necessary to formulate more specific conclusions based on the comments.

Reviewer 2 Report

The manuscript was carefully written and the aim is clear. Taking into consideration the idea of ​​proposing a multisciplinary approach for the management of patients with psoriasis, I recommend inserting a paragraph on genomics.
In particular, this review could be enriched with data relating to pharmacogenomic approaches. I recommend to evaluate the following paper:
- Pharmacogenomics: An Update on Biologics and Small-Molecule Drugs in the Treatment of Psoriasis;
- Influence of Genetic Polymorphisms on Response to Biologics in Moderate-to-Severe Psoriasis.
The material and methods should discuss the search parameters, the reference period and the number of articles discovered, included or excluded.  

Reviewer 3 Report

Beginning with a background on psoriasis and its current mechanistic understanding followed by key biologic treatments available, the authors have reviewed literature to evaluate the connection between biologic treatments for psoriasis and adverse effects. Although the work is thorough, there are some areas where a little more clarity is required. 

Line 73: Did you mean STAT3? Could you please define the abbreviation?

Line 82: The term "alternative" is used but may not be needed here.

Line 96: The sentence about the clinical trial by Gordon et al appears incomplete.

Line 121: Since a replacement to etanercept (adalimumabis mentioned above in Line 104, this portion may be better situated before the mention of adalimubab.

Line 146: In the paragraph starting at this line about Ixekizumab, there are no results for the etanercept groups shown in both UNCOVER-2 and UNCOVER-3 study results even though the text is saying they were performed as comparisons to it and states the number of subjects receiving etanercept in both.

Please recheck sentences from Line 237 to 244, as they seem to lack proper formatting, making them sound confusing.

Line 276: Could the cases of reactivation be random events since it happened in only two cases? Is that enough to suggest there there is a connection?

Line 410: Does this sentence need to be a separate paragraph or could it be included in the following one?

Line 471: How many years of data was included by the authors? Can the findings of this algorithm be briefly summarized?

Line 472: Does the colon suggest that the 2017 study is a part of the 2021 analysis or is it a typo? 

Line 482: Is there a p-value for the risk level comparison?

Discussion: In my opinion, the discussion section should discuss some of the important highlights of your review, which may be supported by other citations. However, in its current form, the discussion appears to be citing findings from new studies that were not mentioned earlier in the text. I think this section could be re-written with a focus on discussing your key findings which may be supported by the currently included text here.

Conclusions: Is there a take-away you would like to include on the risk from biologics or would such a statement be too general to make?

Reviewer 4 Report

The section 3 and 4 should be combined and then divided into subsections based on the scheme 1.

Proposed underlying mechanisms for risks associated with biologic agents in psoriasis should be discussed.

Typing errors should be corrected. For example, line 73 , "STST3" should be corrected as "STAT3".

Round 2

Reviewer 2 Report

the manuscript has been accurately revised.